# Genetic Variants Linked to Myocardial Infarction in Individuals with Non-Alcoholic Fatty Liver Disease and Their Potential Interaction with Dietary Patterns

**DOI:** 10.3390/nu16050602

**Published:** 2024-02-22

**Authors:** Sung-Bum Lee, Ja-Eun Choi, Kyung-Won Hong, Dong-Hyuk Jung

**Affiliations:** 1Department of Family Medicine, Soonchunhyang University Bucheon Hospital, Bucheon 22972, Republic of Korea; sblee@schmc.ac.kr; 2R&D Division, Theragen Health Co., Ltd., Seongnam-si 13493, Republic of Korea; jaeun.choi@theragenhealth.com; 3Department of Family Medicine, Yongin Severance Hospital, Yongin-si 16995, Republic of Korea

**Keywords:** non-alcoholic fatty liver disease, myocardial infarction, GWAS, KoGES

## Abstract

In recent studies, non-alcoholic fatty liver disease (NAFLD) has been associated with a high risk of ischemic heart disease. This study aimed to investigate a genetic variant within a specific gene associated with myocardial infarction (MI) among patients with NAFLD. We included 57,205 participants from a Korean genome and epidemiology study. The baseline population consisted of 45,400 individuals, with 11,805 identified as patients with NAFLD. Genome-wide association studies were conducted for three groups: the entire sample, the healthy population, and patients with NAFLD. We defined the *p*-value < 1 × 10^−5^ as the nominal significance and the *p*-value < 5 × 10^−2^ as statistically significant for the gene-by-nutrient interaction. Among the significant single-nucleotide polymorphisms (SNPs), the lead SNP of each locus was further analyzed. In this cross-sectional study, a total of 1529 participants (2.8%) had experienced MI. Multivariable logistic regression was performed to evaluate the association of 102 SNPs across nine loci. Nine SNPs (rs11891202, rs2278549, rs13146480, rs17293047, rs184257317, rs183081683, rs1887427, rs146939423, and rs76662689) demonstrated an association with MI in the group with NAFLD Notably, the MI-associated SNP, rs134146480, located within the SORCS2 gene, known for its role in secreting insulin in islet cells, showed the most significant association with MI (*p*-value = 2.55 × 10^−7^). Our study identifies candidate genetic polymorphisms associated with NAFLD-related MI. These findings may serve as valuable indicators for estimating MI risk and for conducting future investigations into the underlying mechanisms of NAFLD-related MI.

## 1. Introduction

Non-alcoholic fatty liver disease (NAFLD) is highly prevalent and increasing worldwide, paralleling the recent obesity pandemic. Approximately one in four adults is estimated to be affected by NAFLD [1]. NAFLD encompasses a spectrum of liver injuries caused by excessive ectopic fat accumulation in the liver, without the presence of excessive alcohol consumption or any other apparent liver diseases [2,3,4]. NAFLD could lead to adverse hepatic outcomes, such as liver cirrhosis and hepatocellular carcinoma. Furthermore, it has a detrimental impact on cardiovascular health, including ischemic heart disease and stroke, which are more common causes of death in this population [4,5,6,7]. While the exact causal relationship between NAFLD and cardiovascular disease (CVD) remains unclear, growing evidence suggests that NAFLD is associated with an increased risk of CVD, independent of known CVD risk factors [8,9,10,11].

However, it is important to note that not all patients with NAFLD are necessarily predisposed to adverse CVD outcomes [12]. Identifying the subset of patients with NAFLD predominantly associated with adverse CVD outcomes could help pinpoint specific intervention targets. From this perspective, genetic susceptibility might be one potential target for NAFLD-related CVD. Recent data suggest a possible role of the cumulative effect of multiple common genetic variances, each with an individually small effect size, on CVD risk [13]. Several previous studies have reported the possible pathologic role of genetic variations in coronary artery disease [13,14]. However, the genetic background of NAFLD in the development of myocardial infarction (MI), which could lead to fatal arrhythmia, heart failure, and sudden cardiac death, has not been well characterized. Given this gap in our knowledge, we have designed this study with the objective to investigate a genetic variant within a specific gene that could potentially be associated with MI among patients diagnosed with NAFLD. In other words, the aim of our study was to investigate the genetic association between NAFLD and CVD, which could possibly pave the way for the development of novel therapeutic targets to manage CVD in NAFLD patients. To achieve this goal, we utilized the Korean Genome and Epidemiology Study (KoGES) cohort. This large-scale, comprehensive dataset provides a unique opportunity to explore the interaction between genetic factors and disease outcomes.

## 2. Materials and Methods

### 2.1. Study Design and Population

The Korean Genome and Epidemiology Study (KoGES) is a large prospective cohort study funded by the government, aimed at identifying the genetic and environmental factors contributing to common complex diseases in the Korean population. The cohort consists of community-dwellers, both men and women, aged ≥ 40 years at baseline, who were recruited from the national health examinee registry. The dataset includes anthropometric and clinical measurements, lifestyle information (such as diet, smoking, drinking, and physical activity), and data from a food-frequency questionnaire [15]. In the current study, we included a total of 72,299 participants for whom genome-wide single-nucleotide polymorphism (SNP) data were obtained.

Figure 1 depicts a flow chart that outlines the study protocol. Among the 72,299 participants, 1852 were excluded due to missing values for the assessment of NAFLD. Furthermore, 8920 participants with alcohol consumption ≥140 g per week—which is defined as excessive alcohol intake [16]—and 3721 participants with missing values for alcohol consumption were excluded, as well as 601 participants with missing covariate data. Additionally, 2444 subjects without questionnaire data on a history of MI were excluded. After these exclusions, 54,761 participants were included in the study, with 11,190 of them diagnosed with NAFLD. Finally, we compared 10,745 controls with 445 patients who had experienced MI among the 11,190 NAFLD patients, and we analyzed the significant SNPs associated with MI. This study was approved by the institutional review board (IRB) of Yong-in Severance Hospital on 11 May 2023 (IRB No. 9-2021-0183).

### 2.2. Definitions of NAFLD and MI

NAFLD was defined as hepatic steatosis index (HSI) ≥ 36, which is calculated using the following formula: HSI = 8 × ALT/AST + BMI + SEX (female = 2, male = 0) + DM (yes = 2, no = 0) [17]. MI was defined as the participant-reported history of diagnosis or treatment.

### 2.3. Definitions of Nutrition Intake

To evaluate the dietary intakes of Korean adults in this study, a semi-quantitative food-frequency questionnaire (FFQ) consisting of 103 items was employed for the KoGES [15,18]. The FFQ is a valuable tool commonly used in large population-based studies to investigate the links between dietary habits and chronic diseases. Participants provided information on the frequency and quantity of foods consumed over the past year. The criteria for nutritional intake were determined based on the 2020 Korean Dietary Reference Intakes, which are widely used as reference values for planning and assessing the nutrient intake of individuals.

In the case of macronutrients such as carbohydrates, proteins, and fats, we considered an acceptable macronutrient distribution range (AMDR), representing a range of intakes for energy sources. According to the 2020 Nutrition Reference Intake for Koreans (KDRI), the AMDRs for carbohydrates, proteins, and fats were defined as 55–65%, 7–20%, and 15–30%, respectively [19]. The proportions of nutrient intake were classified as follows in terms of daily total energy intake: carbohydrate intake—high (≥65%) and low (<65%); protein intake—high (≥15%) and low (<15%); fat intake—high (≥20%) and low (<20%).

### 2.4. Genotyping

The genotypes were provided by the Center for Genome Science of the Korea National Institute of Health (KNIH). The genotypes were produced by the Korea Biobank Array (KORV 1.0, Affymetrix, Santa Clara, CA, USA) [20]. The experimental results of the array were filtered by following quality control procedures and criteria: call rate > 97%, minor allele frequency (MAF) > 1%, and Hardy–Weinberg equilibrium test *p*-values > 1 × 10^–5^ [21]. After the quality control procedures, the experiment genotypes were phased using ShapeIT v2, and IMPUTE v2 was used for imputation analyses of the genotype data with 1000 Genomes Phase 3 data for the reference panel. After the imputation, the imputed variants of a quality score of <0.4 or MAF < 1% were excluded from further analyses. The total number of SNPs for the genome-wide association studies (GWASs) was 7,975,321 from chromosomes 1 to 22.

### 2.5. Statistical Analysis

Data were presented as mean ± standard deviation or number with percentages. To compare participants with and without NAFLD, we used Student’s t-tests for continuous variables and Chi-squared tests for categorical variables. To mitigate bias in genomic data arising from variations in the sample collection region, we performed Principal Component Analysis (PCA) [21]. The first two principal components (PC1 and PC2) were included as covariates in the statistical analysis. PC1 and PC2 were derived by using principal component analysis to mitigate the influence of regional differences in the sample collection of genomic data, thereby decreasing the bias. Logistic regression was applied to investigate the genetic variants associated with NAFLD, with adjustments for age, sex, drinking, smoking, exercise, body mass index, total energy intake, PC1, and PC2. All statistical analyses were carried out using PLINK software (ver. 1.9). We considered the *p*-value of < 1 × 10^−5^ as the nominal significance threshold and the *p*-value of < 5 × 10^−2^ as statistically significant for gene-by-nutrient interaction.

## 3. Results

### 3.1. Baseline Characteristics of the Study Population

A total of 54,761 participants (mean age 54.1 ± 8.3 years, 74.1% female) were included in the current analysis, among whom 11,190 participants (20.4%) were classified as having NAFLD. The group with NAFLD comprised slightly older individuals (54.5 ± 8.0 vs. 53.9 ± 8.3 years) with a higher proportion of males (27.3% vs. 25.5%). Individuals in the group with NAFLD exhibited characteristics of metabolic syndrome, as evidenced by higher body mass index, waist circumference, blood pressure, HOMA-IR, and triglyceride levels. The prevalence of comorbidities, including hypertension, type 2 diabetes, and dyslipidemia, was also found to be higher in the group with NAFLD. Table 1 shows the characteristics of the study population. In total, 1529 cases of MI were identified with a higher prevalence observed in the group with NAFLD compared to the non-NAFLD group (3.98% vs. 2.49%).

### 3.2. Association between Genetic Polymorphisms and MI Based on NAFLD

In this study, a total of three independent GWASs were conducted. First, the GWAS analysis for MI case-control was performed in all samples. In addition, the GWAS analysis for MI case-control was performed in both the group with NAFLD and the non-NAFLD group. A total of 102 SNPs were identified as significantly associated with MI in the group with NAFLD, while no association was found between these SNPs and MI in the non-NAFLD group (Appendix A). Furthermore, we selected a cluster pick region to represent the more reliable association and summarized the top SNPs of that region in Table 2. The first column outlines the top significant SNPs among the associated loci, and the final column shows cluster pick loci that have at least three SNPs associated with MI with a *p*-value < 1 × 10^−5^. Nine SNPs (rs11891202, rs2278549, rs13146480, rs17293047, rs184257317, rs183081683, rs1887427, rs146939423, and rs76662689) were associated with MI only in the group with NAFLD. In detail, eight SNPs (rs11891202, rs13146480, rs17293047, rsrs184257317, rs183081683, rs998538, rs146939423, and rs76662689) were positively associated with MI in the group with NAFLD. On the other hand, rs2278549 was inversely associated with MI in the group with NAFLD. Among these SNPs, rs13146480 shows the most significant *p*-value (2.25 × 10^−7^) and is located in the intron region (Figure 2). The odds ratio (OR) was 1.974, which shows that with the minor allele (T) rs13146480, the prevalence of MI increases by 1.974.

The effect of the interaction of nine SNPs with macronutrients on the risk of MI was further analyzed but no significant interaction was identified. In conclusion, the risk of these SNPs are genetic indicators that affect the risk of MI in NAFLD patients; however, there is no specific interaction with diet (Appendix A).

### 3.3. In Silico Annotation of Linked Genes and Functional Relevance

In silico annotation of the positions of SNPs in gene regions is described in Table 3. Eight SNPs were located in nearby the functional gene regions, and one SNP, rs184257317, was located in the intergenic region. The most significant SNP, rs13146480, was located downstream of the sortilin-related VPS 10 domain-containing receptor 2 (SORCS2) gene in the group with NAFLD. Interestingly, the SORCS2 is strongly expressed in sortilin-related VPS10 domain-containing receptor 2. In silico annotation of the SNP function showed that rs183081683 is located in the intron region of the TSPAN12 gene and was reported in the regulation of cell development, activation, growth, and motility. Among the SNPs, rs1887427 is involved in a specific subset of cytokine receptor signaling pathways.

## 4. Discussion

In this study, we found the SNPs potentially associated with NAFLD-related MI in a Korean population. Our study is the first to explore the genetic background of NAFLD-related MI. It shows genetic predispositions may affect the prevalence of MI according to the presence of NAFLD. A total of nine SNPs (rs1891202, rs2278549, rs13146480, rs17293047, rs184257317, rs183081683, rs1887427, rs146939423, and rs76662689) were found to be genetically associated with the prevalence of MI in NAFLD patients. A total of eight SNPs (rs1891202, rs13146480, rs17293047, rs184257317, rs183081683, rs1887427, rs146939423, and rs76662689) containing a minor allele have a positive correlation with MI. On the other hand, rs2278549—containing a minor allele—was negatively associated with MI. Especially, we are able to infer the putative mechanism for two SNPs (rs13146480 and rs1887427), which are two of the most significant SNPs. Even though it is demanding to reveal a direct genetic connection between NAFLD and MI, the presumed mechanism can be found by tracking the mediators linking NAFLD and MI.

An intriguing discovery has been made regarding the first SNP (rs13146480) within the SORCS2 (sortilin-related VPS 10 domain-containing receptor 2) gene, where a minor allele is positively related to myocardial infarction in the group with NAFLD. The gene with a minor allele appears to have a positive association with MI in the group with NAFLD. The SORCS2 gene is particularly interesting because of the role it plays in our cells. VPS 10 domain receptors expressed by the gene act as cargo proteins between the cell surface and intracellular parts, thereby defining secretory and endocytic capacities in cells [22]. SORCS2 is expressed in pancreatic islet cells, which are responsible for the secretion of vital hormones (e.g., insulin and glucagon). If expression of the receptor fails, insulin release from islet beta cells cannot be produced [23]. This suggests an essential implication: mutation of the SORCS2 gene is genetically related to diabetes. It can be inferred that NAFLD is also genetically associated with the SORCS2 gene through the mediator, diabetes mellitus because fatty liver is closely related to diabetes [24]. Furthermore, the gene is also associated with MI because diabetes is a well-known indicator of myocardial infarction [25]. Based upon the pathogenesis, we found rs1314640 with a minor allele is frequent in the group with NAFLD; moreover, it is also genetically associated with the prevalence of myocardial infarction through diabetes.

In the current study, rs1887427 in the janus kinase (JAK) 2 gene was also related to MI in the group with NAFLD. This is an interesting association that deserves further exploration. A previous study found that interruption of JAK2 can lead to disrupting the signaling of growth hormone (GH) in hepatocytes. This interruption, in turn, can cause the occurrence of fatty liver. In other words, deletion of JAK2 can cause fatty liver through hepatocyte-specific growth hormone insensitivity [26,27]. Moreover, inhibition of JAK2 has been shown to reverse and prevent vasoconstriction in arteries, which suggests targeting JAK2 may be a therapeutic strategy for hypertension [28]. JAK2 is also known for mediating intracellular increases in reactive oxygen species, which can remove endothelial nitric oxide, thereby vascular smooth muscle cell contraction [29]. Even more, previous studies found AG 490, widely recognized as an inhibitor of JAK2, was proven to prevent hypertension [30,31]. This finding shows that manipulating JAK2’s activity could be a potential therapeutic strategy for managing hypertension, which is a well-known indicator of MI.

Although a large-scale GWAS study was conducted, there are some limitations. We analyzed the genetic association of NAFLD with the prevalence of MI, not incidence. Consequently, we cannot prove the causality. Genetic associations between MI and NAFLD can only be observed. Another limitation was the demographic content of our study participants. All the participants recruited in the study were Koreans. Further studies including worldwide GWAS data are needed. Third, medication histories such as diabetes, hypertension, and dyslipidemia were not adjusted for in our analysis, which is strongly associated with MIs because they were absent in the data. Moreover, a medication history of hepatitis as well as a past history of HBV or HCV, which may have affected NAFLD, were also not reflected because the data were lacking. Fourth, the gold standard to define NAFLD is a liver biopsy, while imaging studies like MRI and ultrasonography can also be used as diagnostic tools. Nevertheless, because they are intrusive and expensive, these tools were not available in the large cohort study. In this regard, HSI has been utilized in order to define NAFLD. Furthermore, MI was defined according to a participant-answered questionnaire, which might have neglected patients who overlooked their disease.

However, our research has several strengths. This study utilized nationwide data, including a large number of participants (about 60,000). Several studies have previously suggested that some genes were genetically associated with metabolic disease [32,33]. On the other hand, we found gene-related SNPs that are more vulnerable to MI, especially in the group with NAFLD. Further prospective studies are needed to employ these SNPs as indicators of MI.

## 5. Conclusions

In summary, we found nine SNPs were genetically associated with MI in the group with NAFLD. The genetically related SNPs were rs11891202, rs2278549, rs13146480, rs17293047, rs184257317, rs183081683, rs1887427, rs46939423, and rs76662689. According to the type of allele in these SNPs, the prevalence of MI increased in patients with NAFLD. Consequently, the grasp of individuals’ genotyping is important to detect their comorbidities earlier.

## Figures and Tables

**Figure 1 nutrients-16-00602-f001:**
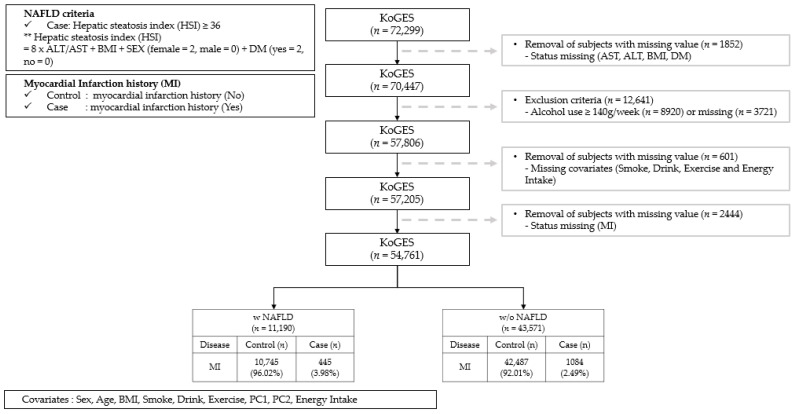
Flowchart for the selection of study participants.

**Figure 2 nutrients-16-00602-f002:**
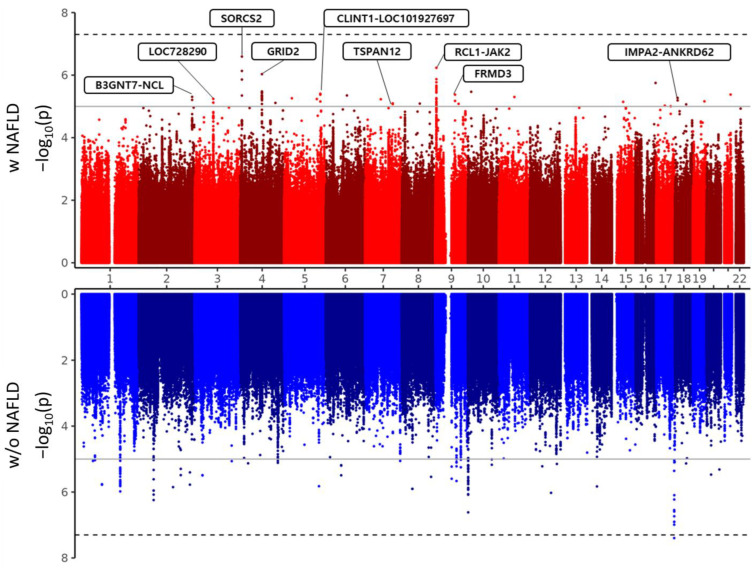
Miami plots represent all p-values for each SNP derived from GWAS analysis as -log transformed p-values. Red plot are the case-control test results for MI within the group with high NAFLD risk (w NAFLD), and blue plots are the case-control results for MI within the group with low NAFLD risk (w/o NAFLD).

**Table 1 nutrients-16-00602-t001:** Baseline characteristics of the study population according to NAFLD.

Variables	Total	w NAFLD	w/o NAFLD	*p*
*n*	54,761	11,190	43,571	
Female (*n*, %)	40,582 (74.11)	8135 (72.7)	32,447 (74.47)	
Age (mean ± sd)	54.05 ± 8.26	54.54 ± 8.0	53.92 ± 8.32	<0.001
Alcohol Intake (g/week)	13.48 ± 27.09	12.94 ± 27.0	13.62 ± 27.11	0.017
Myocardial Infarction History (*n*, %)	53,232 (97.21)/1529 (2.79)	10,745 (96.02)/445 (3.98)	42,487 (97.51)/1084 (2.49)	<0.001
Anthropometric traits				
Body mass index (kg/m^2^)	23.89 ± 2.92	27.43 ± 2.59	22.98 ± 2.23	<0.001
Waist circumference (cm)	80.38 ± 8.61	88.52 ± 7.43	78.29 ± 7.59	<0.001
Systolic blood pressure (mmHg)	121.46 ± 15.31	126.37 ± 15.2	120.19 ± 15.08	<0.001
Diastolic blood pressure (mmHg)	75.23 ± 9.88	78.33 ± 9.82	74.44 ± 9.73	<0.001
Biochemical traits				
Fasting plasma glucose (mg/dl)	94.21 ± 19.01	102.42 ± 26.46	92.11 ± 15.9	<0.001
HbA1c	5.71 ± 0.73	6.09 ± 1.03	5.62 ± 0.6	<0.001
hs-CRP	0.25 ± 1.22	0.35 ± 1.33	0.22 ± 1.19	<0.001
INSUL_FBS	7.74 ± 4.86	9.44 ± 4.68	7.15 ± 4.78	<0.001
HOMA-IR	1.3 ± 1.22	1.81 ± 1.41	1.12 ± 1.1	<0.001
Total cholesterol (mg/dl)	198.06 ± 35.65	202.59 ± 38.24	196.89 ± 34.86	<0.001
HDL cholesterol (mg/dl)	53.11 ± 13.09	48.5 ± 11.31	54.29 ± 13.25	<0.001
Triglyceride (mg/dl)	123.25 ± 80.29	157.06 ± 101.67	114.56 ± 71.25	<0.001
r-glutamyltransferase	25.5 ± 27.01	36.72 ± 35.5	22.63 ± 23.52	<0.001
AST	23.52 ± 22.61	27.48 ± 45.23	22.51 ± 10.59	<0.001
ALT	21.89 ± 22.82	34.75 ± 43.1	18.58 ± 11.15	<0.001
ALP	163.29 ± 106.68	173.78 ± 106.66	160.73 ± 106.53	<0.001
Albumin	4.56 ± 0.47	4.55 ± 0.54	4.56 ± 0.45	0.020
Blood Urea Nitrogen	14.46 ± 4.03	14.8 ± 4.04	14.37 ± 4.03	<0.001
Creatinine	0.8 ± 0.2	0.81 ± 0.21	0.8 ± 0.2	<0.001
Uric Acid	4.55 ± 1.2	4.94 ± 1.26	4.46 ± 1.17	<0.001
Disease History				
Hypertension (*n*, %)	10,663 (19.47)	3579 (31.98)	7084 (16.26)	<0.001
Type 2 Diabetes (*n*, %)	3551 (6.48)	1823 (16.29)	1728 (3.97)	<0.001
Dyslipidemia (*n*, %)	5387 (9.84)	1589 (14.2)	3798 (8.72)	<0.001
Nutrient				
Energy (kcal)	1729.34 ± 557.38	1756.92 ± 561.31	1722.26 ± 556.15	<0.001
Protein (%)	13.24 ± 2.52	13.25 ± 2.54	13.24 ± 2.52	0.687
Fat (%)	13.46 ± 5.35	13.27 ± 5.3	13.5 ± 5.36	<0.001
Carbohydrate (%)	72.28 ± 6.87	72.41 ± 6.83	72.25 ± 6.88	0.032
Calcium (mg)	442.38 ± 256.28	443.49 ± 255.15	442.1 ± 256.57	0.606
Phosphorus (mg)	881.69 ± 354.26	895.05 ± 350.59	878.26 ± 355.12	<0.001
Iron (mg)	9.83 ± 4.85	9.92 ± 4.83	9.8 ± 4.85	0.021
Potassium (mg)	2216.09 ± 1041.95	2249.75 ± 1050.08	2207.45 ± 1039.69	<0.001
Vitamin A (R.E)	468.36 ± 336.24	476.29 ± 342.78	466.33 ± 334.51	0.006
Sodium (mg)	2443.04 ± 1382.47	2535.32 ± 1434.55	2419.34 ± 1367.78	<0.001
Vitamin B1 (mg)	0.99 ± 0.44	1.0 ± 0.44	0.98 ± 0.44	<0.001
Vitamin B2 (mg)	0.88 ± 0.44	0.89 ± 0.44	0.88 ± 0.44	0.005
Niacin (mg)	14.13 ± 6.04	14.35 ± 5.92	14.07 ± 6.07	<0.001
Vitamin C (mg)	106.98 ± 70.01	108.21 ± 71.45	106.66 ± 69.63	0.041
Zinc (μg)	7.79 ± 3.57	7.89 ± 3.44	7.76 ± 3.61	<0.001
Vitamin B6 (mg)	1.56 ± 0.68	1.59 ± 0.68	1.56 ± 0.68	<0.001
Folate (μg)	215.48 ± 119.14	217.69 ± 120.41	214.91 ± 118.81	0.029
Fiber (g)	5.75 ± 2.88	5.84 ± 2.94	5.73 ± 2.86	<0.001
Vitamin E (mg)	8.02 ± 4.47	8.13 ± 4.57	7.99 ± 4.45	<0.001
Cholesterol (mg)	163.12 ± 123.01	161.78 ± 118.25	163.46 ± 124.2	0.184
Lifestyle				
Drinking status: Never/Quit/Current (*n*, %)	34,071 (62.22)/78 (0.14)/20,612 (37.64)	7198 (64.33)/26 (0.23)/3966 (35.44)	26,873 (61.68)/52 (0.12)/16,646 (38.2)	<0.001
Smoking status: Never/Quit/Current (*n*, %)	44,283 (80.87)/6123 (11.18)/4355 (7.95)	8888 (79.43)/1276 (11.4)/1026 (9.17)	35,395 (81.24)/4847 (11.12)/3329 (7.64)	<0.001
Exercise status: No/Yes (*n*, %)	26,984 (49.28)/27,777 (50.72)	5974 (53.39)/5216 (46.61)	21,010 (48.22)/22,561 (51.78)	<0.001
Macronutrient Intake				
Protein (Low/High)	45,560 (79.64)/11,645 (20.36)	9364 (79.32)/2441 (20.68)	36,196 (79.73)/9204 (20.27)	0.421
Carbohydrate (Low/High)	7680 (13.43)/49,525 (86.57)	1573 (13.32)/10,232 (86.68)	6107 (13.45)/39,293 (86.55)	0.650
Fat (Low/High)	50,968 (89.1)/6237 (10.9)	10,559 (89.45)/1246 (10.55)	40,409 (89.01)/4991 (10.99)	0.183
High-Carbohydrate–Low-Protein	44,163 (87.55)	9093 (87.47)	35,070 (87.56)	0.880
Low-Carbohydrate–High-Protein	6283 (12.45)	1302 (12.53)	4981 (12.44)	

NAFLD: non-alcoholic fatty liver disease; w: with; w/o: without; INSUL_FBS: serum insulin in fasting state; AST: aspartate aminotransferase; ALT: alanine aminotransferase; ALP: alkaline phosphatase. All data are presented as mean ± SD, proportion, or median (interquartile range) for skewed variables. *p*-values were calculated by *t*-test for continuous variables and chi-squared test for categorical variables.

**Table 2 nutrients-16-00602-t002:** Top significant SNPs of ischemic heart disease case-control GWASs in the association cluster locus.

SNP	CHR	BP	Feature	Gene	ALT	REF	1KG Phase1 ALT freq.			A1	Myocardial Infraction History (MI)					
							EAS	EUR	AMR		Total		w NAFLD		Control	
											OR (95% CI)	*p*	OR (95% CI)	*p*	OR (95% CI)	*p*
rs11891202	2	232276827	intergenic_region	B3GNT7-NCL	T	C	0.197	0.144	0.114	T	1.15 (1.06–1.26)	1.49 × 10^−3^	1.44 (1.23–1.68)	4.95 × 10^−6^	1.05 (0.94–1.16)	3.94 × 10^−1^
rs2278549	3	81144666	non_coding_transcript_exon_variant	LOC728290	G	T	0.396	0.100	0.138	G	0.93 (0.86–1.01)	6.56 × 10^−2^	0.70 (0.61–0.82)	5.74 × 10^−6^	1.04 (0.95–1.14)	4.43 × 10^−1^
rs13146480	4	7483798	intron_variant	SORCS2	C	T	0.044	0.577	0.484	T	1.24 (1.05–1.47)	1.11 × 10^−2^	1.97 (1.52–2.56)	2.55 × 10^−7^	0.94 (0.76–1.18)	6.01 × 10^−1^
rs17293047	4	94588105	intron_variant	GRID2	A	G	0.073	0.128	0.068	A	1.37 (1.11–1.68)	2.72 × 10^−3^	2.20 (1.60–3.01)	9.30 × 10^−7^	1.03 (0.78–1.36)	8.24 × 10^−1^
rs184257317	5	157464382	intergenic_region	CLINT1-LOC101927697	T	C	0.051	0.000	0.027	T	1.31 (1.09–1.59)	4.92 × 10^−3^	2.04 (1.51–2.77)	3.93 × 10^−6^	1.05 (0.82–1.34)	7.20 × 10^−1^
rs183081683	7	120478350	intron_variant	TSPAN12	A	C	0.019	0.000	0.000	A	1.20 (0.97–1.50)	9.39 × 10^−2^	2.09 (1.51–2.89)	7.98 × 10^−6^	0.86 (0.64–1.16)	3.19 × 10^−1^
rs1887427	9	4979730	intergenic_region	RCL1-JAK2	G	A	0.162	0.259	0.193	G	1.13 (1.03–1.24)	1.07 × 10^−2^	1.51 (1.28–1.77)	5.83 × 10^−7^	0.99 (0.88–1.11)	8.71 × 10^−1^
rs146939423	9	86095179	intron_variant	FRMD3	G	A	0.039	0.000	0.000	G	1.24 (1.01–1.52)	4.19 × 10^−2^	2.09 (1.53–2.86)	4.20 × 10^−6^	0.92 (0.70–1.21)	5.64 × 10^−1^
rs76662689	18	12079732	intergenic_region	IMPA2-ANKRD62	A	G	0.034	0.000	0.001	A	1.42 (1.11–1.81)	5.34 × 10^−3^	2.48 (1.68–3.67)	5.35 × 10^−6^	1.08 (0.78–1.49)	6.52 × 10^−1^

SNP: single-nucleotide polymorphism; CHR: chromosome; BP: base pair; Feature: functional consequence of the SNP position; Gene: nearby functional gene of the SNP position; ALT: alternative allele; REF: reference allele; 1KG Phase 1 ALT freq.: the ratio of frequencies of alternative alleles identified in the 1000 genomes database phase 1 dataset; EAS: East Asian samples; EUR: European samples; AMR: American samples; NAFLD: non-alcoholic fatty liver disease; OR: odds ratio; CI: confidence interval; *p*: *p*-value.

**Table 3 nutrients-16-00602-t003:** Interpretations of significant SNPs.

SNP	CHR	BP	Nearby Genes	Description	Expected Function
rs11891202	2	232276827	B3GNT7	UDP-GlcNAc:betaGal beta-1,3-N-acetylglucosaminyltransferase 7	Preventing cells from migrating out of the original tissues and invading surrounding tissues.
			NCL	nucleolin	It induces chromatin decondensation by binding to histone H1. It is thought to play a role in pre-rRNA transcription and ribosome assembly.
rs2278549	3	81144666	LINC02027	long intergenic non-protein coding RNA 2027	Not known
rs13146480	4	7483798	SORCS2	sortilin-related VPS10 domain-containing receptor 2	These genes are strongly expressed in the central nervous system
rs17293047	4	94588105	GRID2	glutamate ionotropic receptor delta type subunit 2	Synapse organization between parallel fibers and Purkinje cells.
rs184257317	5	157464382	5q33.3	Intergenic	
rs183081683	7	120478350	TSPAN12	tetraspanin 12	The regulation of cell development, activation, growth, and motility.
rs1887427	9	4979730	JAK2	tyrosine-protein kinase JAK2	Involved in a specific subset of cytokine receptor signaling pathways.
rs146939423	9	86095179	FRMD3	FERM domain containing 3	Determine the shape of red blood cells and the function of the encoded protein.
rs76662689	18	12079732	ANKRD62	ankyrin repeat domain 62	Not known

## Data Availability

The genome study used the dataset originally generated in the KoGES supported by the Korean National Institute of Health (KNIH). The subject information and SNP genotype data used in this study are owned entirely by the KNIH, and disclosure of the raw data to the public without permission is strictly prohibited. In principle, the raw data of subject information and SNP genotype used in this study are available with permission from the Institutional Review Board of KNIH for researchers in Korea who meet confidential data access criteria. These data can also be available for researchers overseas when undertaking an international cooperative research project and when the KNIH approves it.

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
