# Peer review of "Genetic Variants Linked to Myocardial Infarction in Individuals with Non-Alcoholic Fatty Liver Disease and Their Potential Interaction with Dietary Patterns"

_nutrients, 2024, doi:10.3390/nu16050602_

Round 1
Reviewer 1 Report
Comments and Suggestions for Authors
Dear Authors.
The relationship between NAFLD and coronary heart disease has been proven, but there is little work on the mechanisms linking these two entities. It seems that the main one is obesity. In your article, you tackled a very important topic assessing the impact of genetic variants on this relationship. In my opinion, the work is very interesting and worth publishing, but it requires a few explanations and additions. I posted my comments below:
1. In the introduction:
- line 49 - incorrect citation record - it is (10) and it should be [10].
- it is worth providing more information about the relationship between NAFLD and chronic coronary syndrome and MI, with particular emphasis on independently of established risk factors (e.g. Diabetes Metab Syndr. 2024 Jan 3;18(1):102938, Life (Basel). 2022 Aug 4;12(8):1189, Curr Probl Cardiol. 2023 Jun;48(6):101643, J Gastroenterol Hepatol. 2020 May;35(5):833-839)
2. Materials and methods:
- the diagnosis of NAFLD was based on HSI. It is not an ideal tool for diagnosing NAFLD, but it can be used on a wide scale. It is worth including this in the discussion in the limitations section of your study.
- why did you specify the value of >140g of alcohol per week as a criterion for excluding NAFLD? what were you guided by - maybe it is worth providing the literature here?
- it is worth writing here that apart from alcohol, other causes of fatty liver disease have also been ruled out (medications, HCV, HBV...)
- I don't understand why you included data on nutrition intake in this paper? I am surprised by such data and their connection with the topic of the work? maybe there is some logical explanation but it is not included in the work. In my opinion, these are very interesting data, although completely unnecessary in this work.
3. Results
- very sparse description of the abbreviations used in table 1 (e.g. what do the letters w and w/o mean in NAFLD? you need to expand all the abbreviations under the table so that there is no doubt about what you are describing in it).
- I don't understand the description of table 2 (what is 1KG Phase1 ALT?)
3. Discussion
- are any of the described SNPs associated with non-genetic risk factors for CAD or IM (including obesity, atherogenic dyslipidemia, type 2 diabetes, hypertension)?
I will be happy to read the article again after considering my doubts.
Author Response
Reply to Reviewer #1
We appreciate the reviewer`s helpful comments regarding our manuscript. We have revised the manuscript in accordance with the recommendations, as explained below in a point-by-point manner.
Dear Authors.
The relationship between NAFLD and coronary heart disease has been proven, but there is little work on the mechanisms linking these two entities. It seems that the main one is obesity. In your article, you tackled a very important topic assessing the impact of genetic variants on this relationship. In my opinion, the work is very interesting and worth publishing, but it requires a few explanations and additions. I posted my comments below:
- In the introduction:
- line 49 - incorrect citation record - it is (10) and it should be [10].
Response: In compliance with the reviewer`s comment, () has been modified with [].
In introduction section:
Recent data suggests a possible role of the cumulative effect of multiple common genetic variances, each with an individually small effect size, on CVD risk [13].
- it is worth providing more information about the relationship between NAFLD and chronic coronary syndrome and MI, with particular emphasis on independently of established risk factors (e.g. Diabetes Metab Syndr. 2024 Jan 3;18(1):102938, Life (Basel). 2022 Aug 4;12(8):1189, Curr Probl Cardiol. 2023 Jun;48(6):101643, J Gastroenterol Hepatol. 2020 May;35(5):833-839)
Response: In accordance with the reviewer`s comment, we have added the more recent version of the references.
In introduction section:
While the exact causal relationship between NAFLD and cardiovascular disease (CVD) remains unclear, growing evidence suggests that NAFLD is associated with an increased risk of CVD, independent of known CVD risk factors [8-11].
In references section:
- Abosheaishaa, H.; Hussein, M.; Ghallab, M.; Abdelhamid, M.; Balassiano, N.; Ahammed, M.R.; Baig, M.A.; Khan, J.; Elshair, M.; Soliman, M.Y. Association between non-alcoholic fatty liver disease and coronary artery disease outcomes: A systematic review and meta-analysis. Diabetes & Metabolic Syndrome: Clinical Research & Reviews 2024, 102938.
- Cazac, G.-D.; Lăcătușu, C.-M.; Mihai, C.; Grigorescu, E.-D.; Onofriescu, A.; Mihai, B.-M. New insights into non-alcoholic fatty liver disease and coronary artery disease: The liver-heart axis. Life 2022, 12, 1189.
- Bisaccia, G.; Ricci, F.; Khanji, M.Y.; Sorella, A.; Melchiorre, E.; Iannetti, G.; Galanti, K.; Mantini, C.; Pizzi, A.D.; Tana, C. Cardiovascular morbidity and mortality related to non-alcoholic fatty liver disease: A systematic review and meta-analysis. Current Problems in Cardiology 2023, 48, 101643.
- Sinn, D.H.; Kang, D.; Chang, Y.; Ryu, S.; Cho, S.J.; Paik, S.W.; Song, Y.B.; Pastor‐Barriuso, R.; Guallar, E.; Cho, J. Non‐alcoholic fatty liver disease and the incidence of myocardial infarction: A cohort study. Journal of Gastroenterology and Hepatology 2020, 35, 833-839.
- Materials and methods:
- the diagnosis of NAFLD was based on HSI. It is not an ideal tool for diagnosing NAFLD, but it can be used on a wide scale. It is worth including this in the discussion in the limitations section of your study.
Response: As the reviewer commented, the gold standard in order to define NAFLD is a liver biopsy. Imaging studies (e.g., MRI or ultrasonography) could also be diagnostic tools. Nevertheless, these tools were not available in the large scaled cohort study. In this regard, HSI has been utilized in order to define NAFLD (Chen, Mao et al. 2023). We have added the information in the discussion section as limitation.
In discussion section:
Fourth, the gold standard to define NAFLD is a liver biopsy, while imaging studies like MRI and ultrasonography can also be used as diagnostic tools. Nevertheless, because they are intrusive and expensive these tools were not available in the large cohort study. In this regard, HSI has been utilized in order to define NAFLD. Furthermore, MI was defined according to a participant-answered questionnaire. It might have neglected patients who overlooked their disease.
Reference for the reviewer:
- Chen, J., X. Mao, M. Deng and G. Luo (2023). "Validation of nonalcoholic fatty liver disease (NAFLD) related steatosis indices in metabolic associated fatty liver disease (MAFLD) and comparison of the diagnostic accuracy between NAFLD and MAFLD." European Journal of Gastroenterology & Hepatology 35(4): 394.
- why did you specify the value of >140g of alcohol per week as a criterion for excluding NAFLD? what were you guided by - maybe it is worth providing the literature here?
Response: Bessembinders et al. defined excessive alcohol intake as over 140g per week. Based on the definition, the participants with excessive alcohol intake were excluded in our study, which can impact on NAFLD. We have added the information with the reference in method section.
In method section / study design and population subsection:
Furthermore, 8,920 participants with alcohol consumption ≥ 140g per week, which is defined as excessive alcohol intake [16].
In references section:
- Bessembinders, K.; Wielders, J.; van de Wiel, A. Severe hypertriglyceridemia influenced by alcohol (shiba). Alcohol and alcoholism 2011, 46, 113-116.
- it is worth writing here that apart from alcohol, other causes of fatty liver disease have also been ruled out (medications, HCV, HBV...)
Response: The cohort data do not provide detailed information on HCV or HBV or medication other than hypertension, diabetes, and hyperlipidemia. We have added the information in the discussion section as limitation.
In discussion section:
Third, medication histories such as diabetes, hypertension and dyslipidemia were not adjusted in our analysis, which is strongly associated with the MI because they were ab-sent for the data. Moreover, Medication history of hepatitis as well as past history of HBV or HCV which may have affected NAFLD were not also reflected because the data is lacking.
- I don't understand why you included data on nutrition intake in this paper? I am surprised by such data and their connection with the topic of the work? maybe there is some logical explanation but it is not included in the work. In my opinion, these are very interesting data, although completely unnecessary in this work.
Response: We appreciate the reviewer`s sharp comment. In compliance with the special issue “Dietary patterns and cardiovascular disease”, we have analyzed the genetic association of dietary pattern with MI in NAFLD patients. While we did not find statistical significance for dietary patterns, these results are also meaningful, thereby including them in Supplementary Table 2.
- Results
- very sparse description of the abbreviations used in table 1 (e.g. what do the letters w and w/o mean in NAFLD? you need to expand all the abbreviations under the table so that there is no doubt about what you are describing in it).
Response: As the reviewer commented, we have added the abbreviations list in table 1.
In result section / table 1:
NAFLD: non-alcoholic fatty liver disease; w: with; w/o: without; INSUL_FBS: serum insulin in fasting state; AST: aspartate aminotransferase; ALT: alanine aminotransferase; ALP: alkaline phosphatase; All data are presented as mean ± SD, proportion, or median (interquartile range) for skewed variables. P-values were calculated by t-test for continuous variables and chi-square test for categorical variables.
- I don't understand the description of table 2 (what is 1KG Phase1 ALT?)
Response: We appreciate the reviewer`s thorough comment. It means alternative allele, which is identified in the 1000 Genome Database Phase 1 dataset. We have added the information with abbreviation list in table 2.
In result section / table 2:
SNP: Single Nucleotide Polymorphism; CHR: Chromosome; BP: Base Pair; Feature: Functional Consequence of the SNP position; Gene: Nearby Functional Gene of the SNP position; ALT: Alternative Allele; REF: Reference Allele; 1KG Phase 1 ALT freq.: the ratio of frequencies of alternative alleles identified in the 1000 genomes Database phase 1 dataset; EAS: East Asian Samples; EUR: European Samples; AMR: American Samples; NAFLD: Non-Alcoholic Fatty Liver Disease; OR: Odds Ratio; CI: Confidence Interval; p: p-value
- Discussion
- are any of the described SNPs associated with non-genetic risk factors for CAD or IM (including obesity, atherogenic dyslipidemia, type 2 diabetes, hypertension)?
Response: Albeit extensive investigation, the association of the addressed SNPs (rs11891202, rs2278549, rs13146480, rs17293047, rs184257317, rs183081683, rs146939423, rs1887427, and rs76662689) with non-genetic risk factors for CAD or IM such as obesity, dyslipidemia, diabetes, and hypertension has not been found in any published research.

Reviewer 2 Report
Comments and Suggestions for Authors
The topic of the relationship between myocardial infarction and nonalcoholic fatty liver disease is of great interest to Nutrients readers. Equally interesting is the relationship with genetic and dietary aspects.
The introduction, although well done, is very short and should be expanded to focus the article's subject.
The methodology is adequate to achieve the proposed objectives.
The results are well presented in the tables and figures.
The discussion is also correct although more bibliographic references to previous studies by other authors are missing.
The bibliographic references used are correct although somewhat obsolete since the obsolescence index (median age of citations) is over 9 years, only 25% are less than 3 years old and just over 7% are less than 1 year old.
I recommend :
Expand the introduction and discussion
Update the citations
Comments on the limitations of the study are missing.
Author Response
Reply to Reviewer #2
We are grateful for the reviewer`s practical comments regarding our manuscript. We have modified the manuscript in compliance with the suggestions, as explained below in a point-by-point manner.
The topic of the relationship between myocardial infarction and nonalcoholic fatty liver disease is of great interest to Nutrients readers. Equally interesting is the relationship with genetic and dietary aspects.
The introduction, although well done, is very short and should be expanded to focus the article's subject.
The methodology is adequate to achieve the proposed objectives.
The results are well presented in the tables and figures.
The discussion is also correct although more bibliographic references to previous studies by other authors are missing.
The bibliographic references used are correct although somewhat obsolete since the obsolescence index (median age of citations) is over 9 years, only 25% are less than 3 years old and just over 7% are less than 1 year old.
I recommend :
Expand the introduction and discussion
Response: As the reviewer`s recommendation, contents of the introduction and discussion has been expanded.
In introduction section:
However, the genetic background of NAFLD in the development of myocardial infarction (MI), which could lead to fatal arrhythmia, heart failure, and sudden cardiac death, has not been well characterized. Accordingly, we designed this study to investigate a genetic variant within a specific gene associated with MI among patients with NAFLD using the Korean Genome and Epidemiology Study (KoGES) cohort. Given this gap in our knowledge, we have designed this study with the objective to investigate a genetic variant within a specific gene that could potentially be associated with MI among patients diagnosed with NAFLD. In other words, the aim of our study was to investigate the genetic association between NAFLD and CVD, which could possibly pave the way for the development of novel therapeutic target to manage CVD in NAFLD patients. To achieve the goal, we have utilized the Korean Genome and Epidemiology Study (KoGES) cohort. This large-scale, comprehensive dataset provides a unique opportunity to explore the interaction between genetic factors and disease outcomes.
In discussion section:
Third, medication histories such as diabetes, hypertension and dyslipidemia were not adjusted in our analysis, which is strongly associated with the MI because they were absent for the data. Moreover, medication history of hepatitis as well as past history of HBV or HCV which may have affected NAFLD was not also reflected because the data is lacking. Fourth, the gold standard to define NAFLD is a liver biopsy, while imaging studies like MRI and ultrasonography can also be used as diagnostic tools. Nevertheless, because they are intrusive and expensive these tools were not available in the large cohort study. In this regard, HSI has been utilized in order to define NAFLD. Furthermore, MI was defined according to a participant-answered questionnaire. It might have neglected patients who overlooked their disease.
Update the citations
Response: As the reviewer`s recommendation, 8 papers published before the 2010s have been replaced with more recent ones. Moreover, we have changed some sentences in order to take into account the modified references.
In discussion section:
This interruption, in turn, can cause the occurrence of fatty liver. In other words, deletion of JAK2 can cause fatty liver through hepatocyte-specific growth hormone insensitivity [26,27]. Moreover, inhibition of JAK2 has been shown to reverse and prevent vasoconstriction in arteries, which suggests targeting JAK2 may be a therapeutic strategy for hypertension [28].
In references section:
- Bali, A.D.; Rosenzveig, A.; Frishman, W.H.; Aronow, W.S. Nonalcoholic fatty liver disease and cardiovascular disease: Causation or association. Cardiology in Review 2023, e000537.
- Chen, J.; Mao, X.; Deng, M.; Luo, G. Validation of nonalcoholic fatty liver disease (nafld) related steatosis indices in metabolic associated fatty liver disease (mafld) and comparison of the diagnostic accuracy between nafld and mafld. European Journal of Gastroenterology & Hepatology 2023, 35, 394.
- Kim, S.; Chung, J.; Ahn, S.; Joung, H. Development of semi-quantitative food frequency questionnaire for obese korean adults. Proceedings of the Nutrition Society 2022, 81, E214.
- Møller, P.L.; Rohde, P.D.; Winther, S.; Breining, P.; Nissen, L.; Nykjaer, A.; Bøttcher, M.; Nyegaard, M.; Kjolby, M. Sortilin as a biomarker for cardiovascular disease revisited. Frontiers in Cardiovascular Medicine 2021, 8, 652584.
- Leopold, J.A. Inhibiting jak2 ameliorates pulmonary hypertension: Fulfilling the promise of precision medicine. American Thoracic Society: 2021; Vol. 64; pp 12-13.
- Kirabo, A.; Kearns, P.N.; Jarajapu, Y.P.; Sasser, J.M.; Oh, S.P.; Grant, M.B.; Kasahara, H.; Cardounel, A.J.; Baylis, C.; Wagner, K.-U. Vascular smooth muscle jak2 mediates angiotensin ii-induced hypertension via increased levels of reactive oxygen species. Cardiovascular research 2011, 91, 171-179.
- Guilluy, C.; Brégeon, J.; Toumaniantz, G.; Rolli-Derkinderen, M.; Retailleau, K.; Loufrani, L.; Henrion, D.; Scalbert, E.; Bril, A.; Torres, R.M. The rho exchange factor arhgef1 mediates the effects of angiotensin ii on vascular tone and blood pressure. Nature medicine 2010, 16, 183-190.
Previous references:
Comments on the limitations of the study are missing.
We appreciate the reviewer`s comment. We have reinforced the limitations in discussion section.
In discussion section:
Albeit a large-scale GWAS study was conducted, there are some limitations. We analyzed the genetic association of NAFLD with the prevalence of MI, not incidence. Consequently, we can`t prove the causality. Genetic association between MI and NAFLD can only be observed. Another limitation was the demographic contents of our study participants. All the participants recruited from the study were Koreans. Further studies including the world-wide GWAS data are needed. Third, medication histories such as diabetes, hypertension and dyslipidemia were not adjusted in our analysis, which is strongly associated with the MI because they were absent for the data. Moreover, medication history of hepatitis as well as past history of HBV or HCV which may have affected NAFLD was not also reflected because the data is lacking. Fourth, the gold standard to define NAFLD is a liver biopsy, while imaging studies like MRI and ultrasonography can also be used as diagnostic tools. Nevertheless, because they are intrusive and expensive these tools were not available in the large cohort study. In this regard, HSI has been utilized in order to define NAFLD. Furthermore, MI was defined according to a participant-answered questionnaire. It might have neglected patients who overlooked their disease.

Round 2
Reviewer 1 Report
Comments and Suggestions for Authors
Dear Authors,
thank you very much for taking my comments and explanations into account. He feels completely satisfied. I have no further comments on the article.
Reviewer 2 Report
Comments and Suggestions for Authors
Everything ok